# Individual Differences on Wellbeing Indices during the COVID-19 Quarantine in Greece: A National Study

**DOI:** 10.3390/ijerph20247182

**Published:** 2023-12-15

**Authors:** Christos Pezirkianidis, Christina Parpoula, Christina Athanasiades, Katerina Flora, Nikolaos Makris, Despina Moraitou, Georgia Papantoniou, Stephanos Vassilopoulos, Maria Sini, Anastassios Stalikas

**Affiliations:** 1Laboratory of Positive Psychology, Department of Psychology, Panteion University of Social & Political Sciences, Syggrou Ave. 136, 17671 Athens, Greece; mariasinipsy@outlook.com.gr (M.S.); anstal@panteion.gr (A.S.); 2Department of Psychology, Panteion University of Social & Political Sciences, Syggrou Ave. 136, 17671 Athens, Greece; chparpoula@panteion.gr; 3Department of Psychology, Aristotle University of Thessaloniki, 54124 Thessaloniki, Greece; cathan@psy.auth.gr (C.A.); demorait@psy.auth.gr (D.M.); 4Department of Psychology, University of Western Macedonia, 50100 Kozani, Greece; kflora@uowm.gr; 5Department of Primary Education, Democritus University of Thrace, 68100 Alexandroupolis, Greece; nmakris@eled.duth.gr; 6Laboratory of Neurodegenerative Diseases, Center for Interdisciplinary Research and Innovation (CIRI-AUTH), Balkan Center, Aristotle University of Thessaloniki, 10th km Thessaloniki-Thermi, 54124 Thessaloniki, Greece; gpapanto@uoi.gr; 7Laboratory of Psychology, Department of Early Childhood Education, School of Education, University of Ioannina, 45110 Ioannina, Greece; 8Department of Educational Sciences and Social Work, University of Patras, 26110 Patras, Greece; stephanosv@upatras.gr

**Keywords:** wellbeing, individual differences, COVID-19, quarantine, meaning in life, social relationships, depression, anxiety, stress, Greece

## Abstract

The impact of COVID-19 and the associated lockdown measures on people’s physical and mental wellbeing, as well as their daily lives and functioning, has been extensively studied. This study takes the approach of investigating the consequences of COVID-19 on a national scale, considering sociodemographic factors. The main objective is to make a contribution to ongoing research by specifically examining how age, gender, and marital status influence the overall impact of COVID-19 and wellbeing indicators during the second lockdown period that was implemented in response to the COVID-19 pandemic in the Greek population. The study involved a sample of 16,906 individuals of all age groups in Greece who completed an online questionnaire encompassing measurements related to personal wellbeing, the presence and search for meaning in life, positive relationships, as well as symptoms of depression, anxiety, and stress. Additionally, to gauge the levels of the perceived COVID-19-related impact, a valid and reliable scale was developed. The results reveal that a higher perception of COVID-19 consequences is positively associated with psychological symptoms and the search for meaning in life, while being negatively correlated with personal wellbeing and the sense of meaning in life. In terms of individual differences, the findings indicate that unmarried individuals, young adults, and females tend to report higher levels of psychological symptoms, a greater search for meaning in life, and a heightened perception of COVID-19-related impact. These findings are analyzed in depth, and suggestions for potential directions for future research are put forth.

## 1. Introduction

The coronavirus disease, first reported in China in December 2019, was declared a global pandemic by the WHO in March 2020. Its spread was rapid, and governments around the world took strict measures to protect citizens and prevent the collapse of healthcare systems. Such measures were the lockdowns and the restrictions on social contacts, which related to a huge burden on the mental health of the general population [1,2]. 

Several studies investigated possible individual differences in the pandemic effects on people’s mental health. 

Regarding age differences, Birditt et al. [3] found that older adults experienced less stress due to the pandemic, less life change, less social isolation, and higher levels of close relationship quality than younger ones [3]. Similarly, another study found that, compared to younger individuals, older adults showed a lower risk of mental health issues during the COVID-19 pandemic [4].

Moreover, unemployment has been found to be a risk factor for people’s mental health during the pandemic, according to the results of several studies that have investigated the effects of work status [5,6].

In addition, regarding family status, while the existence of a supportive family environment seemed to act as a protective factor for the mental health of individuals during the pandemic [7], some studies found that parents reported increased levels of stress, especially mothers and low-income respondents [8,9].

According to a recent report by the World Health Organization [10], the COVID-19 pandemic had a strong impact on the mental health and wellbeing of people around the globe. Specifically, in 2020, there was a 27.6% increase in cases of major depressive disorder and a 25.6% increase in cases of anxiety disorders. The most vulnerable individuals were women, young people, and people living in low- and middle-income countries [10].

### 1.1. World and National Wellbeing Reports

Global experience has highlighted the importance of creating detailed wellbeing reports that take into account various aspects of humans’ lives, such as life satisfaction and experiencing positive emotions, but also environmental factors, like the wellbeing derived from our home or office structure [11,12,13,14,15,16,17,18]. These reports provide detailed information about the levels of wellbeing of citizens at the national and transnational level, and outline their needs. This information is extremely valuable for the design and implementation of specific interventions that aim to enhance wellbeing at an individual, group, and systemic level, especially during and after the COVID-19 pandemic. On top of that, wellbeing reports significantly contribute to social and public health policymaking [19,20].

The 2021 and 2022 World Happiness Reports focused on the COVID-19 pandemic and its effects on wellbeing. Six factors have been found to crucially relate to wellbeing levels: income, health, someone to count on, freedom, generosity, and trust. Among these, trust in public institutions and income equality were found to play a crucial role in dealing with the COVID-19 pandemic effects [21,22].

Regarding mental health outcomes, an 8% increase in stress and worry in 2021 and a 4% increase in 2022 was reported compared to prepandemic levels. Despite this, the reports also showed a remarkable global increase in benevolence (25% higher), measured by the average of the prosocial behaviors of donations, volunteering, and helping strangers [21,22]. 

Apart from the World Happiness Report, several countries publish annual national wellbeing reports. In Australia, for instance, a national survey taking place regarding the first wave of the COVID-19 pandemic showed that Australians’ wellbeing was not affected overall. Researchers argue that the Australians’ resilience was due to the positive effects of this pandemic, such as spending quality time with family members [23]. European countries are taking the first steps in publishing national wellbeing reports and contact national studies, while Greece is out of this arena to this day.

### 1.2. Indicators of Wellbeing during the COVID-19 Pandemic

#### 1.2.1. Meaning in Life: Presence and Search for Meaning

As Steger [24] (p. 177) points out, “Meaning in life necessarily involves people feeling that their lives matter, making sense of their lives and determining a broader purpose of their lives”. Meaning in life is a crucial indicator of both physical and psychological wellbeing, and research findings indicate its relation to greater satisfaction with life, while it also lessens suffering and can foster functional coping skills and adjustment for people who have experienced trauma [24,25]. 

Researchers of meaning in life distinguish the presence of meaning from the search for meaning, as they are related to several wellbeing outcomes in different ways. The presence of meaning has been found to lead to greater wellbeing at all developmental stages, whereas the search for meaning correlates negatively with wellbeing in specific samples, especially among older adults [26].

According to several studies conducted during the first year of the COVID-19 pandemic, meaning in life was a protective factor against depression and anxiety [27], and it was related to lower states of anxiety and COVID-19-related stress [28], and could therefore predict resilience [29]. In China, Lin [30] found that pre-existing increased levels of meaning in life were correlated with greater life satisfaction, lower levels of depression, anxiety, stress, and negative emotions, and increased prosocial behavior during the outbreak of the pandemic, indicating that it may contribute to the maintenance of a more positive adjustment.

In a study conducted by de Vries et al. [31], it appeared that women, compared to men, reported lower levels of meaning in life during the pandemic. Furthermore, in terms of educational level, the more highly educated individuals reported increased levels of meaning in life during the pandemic, in contrast to the less educated [31]. To our knowledge, there is no study so far that has examined separately the presence and the search for meaning in life over the course of the COVID-19 pandemic.

#### 1.2.2. Positive Social Relationships

Interpersonal relationships are core indicators of wellbeing and resilience. Humans are social by nature, and it is proven that relationships are essential, not only for survival, but also for living a happy and meaningful life [32,33,34]. 

During the COVID-19 pandemic, one of the main measures implemented to protect public health was social and physical distancing [35]. However, the physical connection is vital for humans’ survival and wellbeing. There is evidence that, during the first months of the pandemic, people experienced feelings of longing for touch, with the greatest levels reported by those living alone [36]. In addition, even though people were able to maintain social connections and communicate using other means, like social media, there was an increase in loneliness [37,38], especially for specific subgroups, such as individuals living alone [37,39].

In general, social relationships have been proven to be protective of wellbeing during the COVID-19 pandemic. Studies conducted through the first year of the pandemic showed that the maintenance of high-quality relationships functioned as a protective factor for wellbeing, predicting lower levels of depression. According to these studies, a key component of high-quality relationships is the perceived social support by others [40,41]. These findings were also consistent with those included in the 2021 World Happiness Report. The quality and the quantity of social relationships were found to be protective factors for wellbeing during the pandemic, especially due to the sense of connectedness they offered [21]. 

Studies focusing on subgroups might shed light on how they could probably differ in terms of their sense of loneliness, social relationships, perceived social support, and wellbeing during the pandemic.

### 1.3. The Effects of COVID-19 on Mental Health in Greece

In Greece, the first COVID-19 case was reported on 26 February 2020. The Greek government took immediate measures to protect citizens and prevent the collapse of the public healthcare system. The first lockdown was implemented in the middle of March 2020 and ended gradually in May 2020. 

During that very first lockdown, people experienced high levels of depression, anxiety, and stress [42,43], while young adults experienced low levels of positive affect and reported being worried, mostly about their family members rather than themselves [43]. Also, specific subgroups that faced the harsh effects of the pandemic, such as health professionals, experienced high levels of PTSD and dehumanization [44,45].

Concerning the positive aspects of wellbeing, Demetriou et al. [46] examined positive indicators of wellbeing among the Greek and Cypriot general population during the first COVID-19 wave and found that most participants reported medium to high levels of resilience and hope, with the more educated ones having greater levels of hope. Furthermore, adults over 40 years old, as well as married women and men, adapted easier to the unprecedented quarantine conditions [46]. Another study examined multiple factors that might have affected the life satisfaction of Greek adults during the first wave of COVID-19, and revealed that women were more vulnerable in terms of life satisfaction, while older adults, and especially people over 65 years old, were more satisfied with life [47]. 

Moreover, two studies highlighted the protective effects of specific character strengths on positive wellbeing components during the COVID-19 pandemic. More specifically, it was found that the character strengths of hope and love positively predicted subjective wellbeing, accomplishments, engagement, and positive relationships during the first wave of the pandemic in Greece [48,49]. However, both of them negatively predicted the presence of meaning in life, possibly because of the strict measures of social distancing and the sense of the unknown that dominated the first months of the pandemic in Greece [49].

Finally, regarding the mental health of children and adolescents during the first wave of the pandemic in Greece, there is evidence that it was negatively affected, as reported by the one-third of the parents participating in research conducted by Magklara et al. [50]. Among the risk factors for their psychological health were the social isolation as well as the unemployment of a parent and the conflicts the family members often had [50].

Regarding the second massive lockdown in Greece, which lasted for about 7 months, there are fewer study findings published. Kalaitzaki et al. [51] examined mental health indicators during the first and the second lockdowns and compared the results afterwards. They found that the second lockdown was more adverse for the general population’s mental health. Greek people reported higher levels of perceived stress, PTSD symptoms, and loneliness, and less adaptive coping strategies, as well as lower resilience levels and less perceived social support [51,52]. Moreover, Gournellis and Efstathiou [53] found that, even though the second lockdown was more adverse in Greece, the depression and suicidal ideation levels did not increase compared to the first lockdown, in contrast to anxiety levels, which significantly increased. 

Especially for Greek university students, research findings indicate that their depression, anxiety, and stress levels were higher compared to the first wave of the pandemic [54]. Also, Kokkinos et al. [55] found that the perceived financial, psychological, and academic impact of the COVID-19 pandemic affected university students’ satisfaction with life, while their general mental health played an important role on this relationship. Regarding gender differences, female university students seemed to report higher levels of psychological symptoms and psychological impacts from the second wave of the pandemic compared to male students [54,55].

### 1.4. The Present Study 

The effect of COVID-19 and related lockdown phases on peoples’ physical and mental health, as well as their everyday life and functioning, is a vital component of individual quality of life, which has been explored by researchers from different perspectives during the pandemic. This study takes the stance to investigate the COVID-19-related consequences nationwide from the sociodemographic standpoint. The purpose is to contribute to further research by exploring specifically the role of age, gender, and marital status on the overall COVID-19 impact and wellbeing indices during the second quarantine period imposed due to the COVID-19 pandemic on the Greek population. More specifically, the aim is to examine gender, age, and marital status differences among Greek citizens across the country on their reports of depression, anxiety, stress, the presence and search for meaning in life, positive interpersonal relationships, personal wellbeing, and the perceived COVID-19 impact. An additional aim of the present study is to construct a psychological instrument to measure the impact of the COVID-19 pandemic.

In line with this purpose, answers are sought for the following research questions:

RQ1: Is a short measure of the perceived COVID-19 impact unidimensional, conceptually valid, and reliable?

RQ2: Is there any significant difference in participants’ levels of the presence of meaning in life, personal wellbeing, and positive relationships in terms of age, gender, and marital status?

RQ3: Is there any significant difference in participants’ levels of search for meaning in life and COVID-19-related impact in terms of age, gender, and marital status?

RQ4: Is there any significant difference in participants’ levels of depression, anxiety, and stress in terms of age, gender, and marital status?

The research hypotheses are the following:

**H1:** 
*The perceived COVID-19 impact measure will be unidimensional, reliable, and of high content and construct validity.*


Previous study findings and theories mapped indicators of different aspects of crisis consequences and quality of life, namely, emotional, physical, material, social, and psychological wellbeing, which also can be measured and conceptualized into a higher-order factor context [21,22,25,55,56,57,58]. Moreover, based on previous literature, the COVID-19-related impact will be positively correlated with negative wellbeing indices and negatively correlated to positive wellbeing indices [21,22,28,54,55].

**H2:** 
*Male participants will report significantly higher positive wellbeing indices, lower psychological symptoms, and lower COVID-19 impact compared to female adults.*


Previous literature indicates that women typically report higher levels of negative wellbeing indices than men [59], especially during the pandemic [54,55]. 

**H3:** 
*Middle- and older-age adults will report significantly higher positive wellbeing indices, lower psychological symptoms, and lower COVID-19 impact compared to young adults.*


Previous research suggests that, during the pandemic, older adults reported higher wellbeing [46,47], while young adults reported higher levels of negative wellbeing indices [43,54,55].

**H4:** 
*Married participants and those living with their romantic partner will report significantly higher positive wellbeing indices, lower psychological symptoms, and lower COVID-19 impact compared to single, widowed, and divorced participants.*


Previous studies emphasize the protective role of a supportive family environment for individuals’ mental health during the pandemic [7]. 

## 2. Materials and Methods

### 2.1. Participants

A total of 16,906 Greek individuals of all age groups participated in the study. A total of 65.4% of the participants were women and their mean age was 37 (SD = 15.75). More specifically, 0.7% of the participants were under 17 years of age, 29.4% were aged between 18 and 25, 19.6% between 26 and 35, 15.9% between 36 and 45, 21.3% between 46 and 55, 7.3% were aged between 56 and 65, and 5.7% of them were over 66 years of age (0.1% missing ages). According to the 2014 report from the Hellenic Statistical Authority [60], Greece is home to 49% men and 51% women, with the average age of the population being 41.9 years. Considering that women and individuals with higher education levels (with younger people being more educated in Greece) are more inclined to engage in surveys [61], the sample characteristics in this study closely mirror those of the overall population distribution. 

Regarding their region of residence, most of them reported Central Macedonia (33.1%), Attica (15.7%), Eastern Macedonia and Thrace (10.6%), Epirus (8.4), Western Greece (7.3%), and Thessaly (7.2%). For the rest of the regions (Peloponnese, Crete, Western Macedonia, Central Greece, Ionian Islands, North Aegean, and South Aegean), lower percentages were recorded, ranging from 1.3% to 3.6%.

Regarding their marital status, 47.2% of the participants were unmarried, 44.4% were married or living together, 5.6% were divorced, and 2.8% were widowed. Moreover, 46.3% of them had children, and 24.06% of them reported having one child, 55.29% of them reported having two children, and 20.65% reported having three or more children. Concerning their educational level, most of them were university graduates (32.6%), while 19.7% were university students, 23.8% were high school graduates, 3.8% were middle school graduates, 4.0% were primary school graduates, and 16.2% held a postgraduate degree. According to the 2014 report from the Hellenic Statistical Authority [60], in Greece, 36% of individuals are unmarried, 45% are married, 8% are widowed, and 4% are divorced. Additionally, 20% of the Greek population are university graduates, 20% are high school graduates, 10% are middle school graduates, 23% are primary school graduates, and 2% hold a Master’s or doctoral degree. Consequently, the participants in the current study appear to align with the general population in terms of marital status. However, there were fewer participants with lower educational levels (middle and primary school graduates) and more participants with higher educational levels (postgraduate degrees) compared to the general population in Greece.

Most of the participants worked at the time of the survey (59.9%). More specifically, 14.4% reported to be self-employed, whereas 20% and 25.5% were employees in the public and private sector, respectively. Moreover, 8.1% reported to be unemployed, while 31.8% reported to be university students, perform housekeeping duties, or reported another occupation. Regarding the monthly individual net income, most of the participants (38.4%) reported an income under 600 euros, 21.1% reported an income between 600 and 900 euros, 20.4% between 900 and 1200 euros, 10.6% between 1200 and 1500 euros, 7.5% between 1500 and 3000 euros, and only 2.1% reported an income more than 3000 euros.

At the moment of the study enrollment (December 2020 to April 2021, i.e., the second quarantine due to the COVID-19 pandemic), the vast majority of participants (99.2%) reported that they did not experience COVID-19-related symptoms, while 7% reported that they became sick from COVID-19 in the past. Moreover, almost half of the participants (54.9%) had no close contacts that became sick from COVID-19, 38.6% of the participants reported that a family member or a close friend became sick and recovered, 3.3% reported that a family member or a close friend became seriously ill from COVID-19 and died, and 3.1% of the participants reported that a close contact of them experienced COVID-19 illness at the moment of the study. Regarding their health status, 14.5% of the participants reported belonging to vulnerable and high-risk groups for COVID-19.

Most participants (91.5%) reported no job loss due to the coronavirus pandemic, and 18.8% reported that a close family member had experienced job loss due to the COVID-19 outbreak. Moreover, on a scale from 1 (not at all) to 10 (very much), the participants reported being somewhat worried (M = 4.52, SD = 3.07) that they or a close family member may lose their job in the future.

### 2.2. Measures

#### 2.2.1. Subjective Wellbeing

The Personal Wellbeing Scale [20] has been used by the Australian Association for Wellbeing Index since 2003 to measure personal wellbeing. The scale consists of eight items that measure an individual’s satisfaction with life as a member of the community. Participants answer the questions using a ten-point Likert-type scale (1 = not at all satisfied to 10 = completely satisfied). The scale was adapted to the Greek cultural context using the back-translation procedure as part of this project [62]. The Cronbach’s α in the sample of this survey was adequate (α = 0.86).

#### 2.2.2. Depressive, Anxiety, and Stress Symptoms

The Depression, Anxiety and Stress Scale-21 [63] was used to measure depression, anxiety, and stress symptoms. The scale consists of three subscales that include seven items rated on a four-point Likert-type scale, each to measure the aforementioned negative psychological states. The items are scored on a 10-point Likert-type scale (1 = not at all/never to 10 = absolutely/always). The scale demonstrated adequate psychometric properties in the Greek population [64]. In the present study, the Cronbach’s α reliability index for all three subscales was high (α = 0.91).

#### 2.2.3. Positive Relationships

The 35-item Positive Relationships Questionnaire [65] measures the perceived quality of individuals’ close relationships. The questionnaire measures, specifically, four functional characteristics of positive relationships: practical support, emotional support, self-improvement, and shared enjoyment. The items are scored on a 10-point Likert-type scale (0 = not at all/never to 10 = absolutely/always). A total score can also be extracted. According to the analysis, the Cronbach’s α for the overall scale and the subscales was high (α = 0.98 for total score and α = 0.89 to 0.96 for the four subscales).

#### 2.2.4. Meaning of Life

The Meaning in Life Questionnaire [66] consists of two subscales that measure the existence of meaning and the search for meaning in life. It includes 10 self-report items that are scored on a 10-point Likert-type scale ranging from 1 (completely untrue) to 10 (completely true). The measure has been validated in the Greek population by Pezirkianidis et al. [67] and demonstrated good psychometric properties. In the present study, the Cronbach’s α for each subscale was satisfactory (α = 0.85 for the presence of meaning subscale and α = 0.87 for the search for meaning subscale).

#### 2.2.5. COVID-19 Consequences

We developed the COVID-19 Overall Impact Scale (CV-19OIS), which was composed of six items measuring the overall impact of the COVID-19 pandemic (e.g., “I am annoyed with the changes having occurred in my life due to pandemic” and “Pandemic has psychological impacts on my life”). All items are rated on a 10-point Likert-type scale, ranging from 1 (not at all) to 10 (very much). A total score is calculated by summing up responses. The higher the score, the greater the overall impact of COVID-19, ranging from 6 to 60. The internal consistency of the CV-19OIS was 0.80. 

### 2.3. Procedure

#### 2.3.1. Research Design and Sampling

Data were gathered between December 2020 and April 2021, coinciding with the occurrence of the second quarantine in Greece prompted by the second wave of the COVID-19 pandemic, which endured for seven months. A cross-sectional research approach was employed to portray the current state of wellbeing and mental health among the Greek populace. This study had a nationwide scope, as it benefited from the collaboration of eight Greek universities and marked the inception of the inaugural Greek Wellbeing Observatory, situated within the Laboratory of Positive Psychology at the Panteion University of Social and Political Sciences.

Students hailing from eight diverse Greek universities (namely, Panteion University of Social and Political Sciences, Aristotle University of Thessaloniki, University of Crete, University of Ioannina, University of Thessaly, Democritus University of Thrace, University of Western Macedonia, and University of Ioannina) across various regions (Attica, Thessaly, Central and West Macedonia, Crete, West Greece, Thrace, and Epirus) were provided with training in research design and took on the role of recruiters. Each student from these universities enlisted ten participants from their social milieu, encompassing a range of ages and genders. In particular, the students were tasked with recruiting five men and five women, with four falling within the age bracket of up to 30 years old, three aged up to 45, and three exceeding 46 years of age. Prior to granting their consent, participants were briefed on the study’s objectives and the confidentiality of their input. They were not offered any external incentives or compensation. The data were documented using Google Forms.

#### 2.3.2. Procedure for the Development of the COVID-19 Overall Impact Scale 

One of the aims of the current study was to develop a new scale to examine the overall impact of the COVID-19 pandemic on all aspects of people’s livelihoods, since most studies focused on measuring the impact of COVID-19 on mental health [68]. The CV-19OIS was developed through two separate stages. In the initial stage, we reviewed the literature regarding the impact of COVID-19 on people’s livelihoods, their mental and physical health, employment and labor issues, as well as on the economic and social disruption experienced.

Numerous papers have reported the various effects of COVID-19, but most of these studies were based on data retrieved at the beginning of the pandemic. For this reason, we also reviewed related reports, articles, and announcements published by trusted organizations and sources, such as the World Health Organization, government institutions, major news media, or Google Trends analytics, to investigate its prolonged effects. Based on these reviews, we collected a plethora of reported effects, in the form of keywords, related to everyday life-functioning problems (friendship issues, workplace pressure, traveling, having difficulty readjusting to home or work life, feeling of isolation, marital problems, etc.), physical and emotional distress symptoms (such as headaches, stomach pains, eating or sleeping too much or too little, avoidance, having low or no energy, excessive smoking, drinking, or using drugs, feeling helpless or hopeless, emptiness, worrying a lot of the time; depression, anxiety, anger, suicidal thoughts, irritation, fatigue, etc.), and socioeconomic problems (such as unfair treatment, public health issues, educational issues, workplace issues, job loss, income reduction, and other related impacts and/or concerns) that can be experienced with prolonged pandemic.

In the next stage, we evaluated these keywords (through discussions among the authors), and after removing those with relatively low relevance, we clustered the remaining ones in groups of related content that collectively covered a broad subject area. In this way, key topic clusters emerged, including psychological, physical, social, economic, life-functioning, and daily routine experiences. These key topics also depicted core quality of life indicators (emotional, physical, material, social, and psychological wellbeing) based on prepandemic theories and models [21,22,25,55,56,57,58]. Moreover, these models support that these indicators can be measured separately but also provide information about a latent higher-order variable, which was labeled quality of life, wellbeing, or crisis consequences, respectively. Based on these topic clusters, we generated items describing the subjective impact of the COVID-19 pandemic in significant aspects of everyday life. Each item measured the degree of agreement or distress caused by the COVID-19 pandemic, which was based on a 10-point Likert-type scale ranging from 1 (not at all) to 10 (very much). The generated 7 items were sent to an expert panel (comprising licensed clinical psychologists and professors in clinical and counseling psychology) to obtain the review. After one item was deleted based on the feedback received from the expert panel, the final 6 items were included in the CV-19OIS.

### 2.4. Data Analysis

We first curated the database comprised of 18,011 participants. A total of 1105 cases were excluded after confirming randomness, i.e., participants that answered the highest, lowest, or the same value in all items of the questionnaire. Then, we used descriptive statistics to describe the basic features of the participants. We then examined the psychometric properties of the COVID-19 Overall Impact Scale. We used the whole dataset (*n* = 16,906) to conduct exploratory factor analysis (EFA) exploring the dimensionality of the scale and item loading, adopting a principal axis factoring analysis with oblique rotation. Then, we tested the intercorrelations between the CV-19OIS and positive and negative wellbeing constructs. We expected the CV-19OIS to be positively associated with negative psychological constructs, e.g., depression, anxiety, stress, and the search for meaning in life (convergent validity), and negatively correlated to positive psychological constructs, such as the presence of meaning in life, personal wellbeing, and positive interpersonal relationships (discriminant validity) [69]. Afterward, we tested if the data for each variable significantly deviated from a normal distribution using the Kolmogorov–Smirnov (K-S) test (correcting the K-S for small values at the tails of probability distributions adopting Lilliefors test), and the results showed that all variables do not follow a normal distribution. Thus, nonparametric tests were used for the exploration of independent samples’ differences, comparing the distributions across groups using either the Mann–Whitney U test for two samples or the Kruskal–Wallis one-way ANOVA for more than two samples. All data analyses were conducted using the IBM Statistics SPSS 28 [70].

## 3. Results

### 3.1. Descriptive Statistics and Preliminary Analysis

Table 1 shows the descriptive statistics and internal consistency of the CV-19OIS across the whole sample.

According to Table 2, the average presence of meaning in life (M = 7.22, SD = 1.72), personal wellbeing (M = 7.18, SD = 1.28), and the search for meaning in life (M = 7.14, SD = 4.89) level of the participants was between 7.00 and 8.00 points using a ten-point scale. The mean score of the positive relationships’ variable is one of the highest recorded (M = 8.04, SD = 1.44). The average depression and anxiety levels recorded were moderate (between 3.00 and 4.00 points), and the highest scored DASS-21 subscale/variable recorded was stress (M = 4.64, SD = 2.24), followed by depression (M = 3.87, SD = 2.22) and anxiety (M = 3.41, SD = 2.21). The average COVID-19 overall impact level of the participants was between 6.00 and 7.00 points (M = 6.64, SD = 1.83).

### 3.2. Exploratory Factor Analysis

The whole sample was used for an EFA to examine the factor structure of the CV-19OIS. The significance of Bartlett’s test of sphericity (*χ^2^*(15) = 32150.21, *p* < 0.001) and the Kaiser–Meyer–Olkin measure of sampling adequacy (KMO = 0.84) indicated adequacy of the data for applying the EFA. We conducted the EFA using the principal axis factoring analysis with oblimin (oblique) rotation, which allows factors to be correlated. The number of factors was determined by a combination of the empirical Kaiser criterion [71], the scree plot [72], and the minimum average partial test [73]. The analysis revealed a single factor under a cutoff of an eigenvalue of 1, explaining 52.29% of the total variance. We found that communalities for all items exceeded 0.3 (except item 1), and all items significantly loaded on the factor using exclusion criterion of 0.40. The internal consistency of the CV-19OIS was very good, α = 0.80. Table 3 presents the factor loading and communality of each item.

### 3.3. Correlations

We tested the correlation between the CV-19OIS and other constructs to evaluate its construct validity. The findings showed that the CV-19OIS is positively correlated to depression (r_s_ = 0.35, *p* < 0.001), anxiety (r_s_ = 0.29, *p* < 0.001), stress (r_s_ = 0.36, *p* < 0.001), and the search for meaning in life (r_s_ = 0.08, *p* < 0.001), while it is negatively correlated to personal wellbeing (r_s_ = −0.23, *p* < 0.001) and the presence of meaning in life (r_s_ = −0.12, *p* < 0.001). On the other hand, the CV-19OIS was not significantly correlated with positive interpersonal relationships. Taking everything into account, the newly constructed measure is characterized by adequate convergent and discriminant validity. Moreover, these findings show that higher levels of perceived COVID-19 impact relate to increased psychological symptoms and the search for meaning in life, and a decreased presence of meaning and personal wellbeing.

### 3.4. Individual Differences

#### 3.4.1. Gender Differences

According to Table 4, no significant difference between genders was found in terms of the presence of meaning in life. However, it was observed that males reported statistically significantly higher personal wellbeing than females, whereas the female group reported higher scores on positive relationships. Further, according to Table 5, the group that reported statistically significant higher levels on negative wellbeing indices (i.e., the search for meaning in life, depression, anxiety, and stress), as well as on COVID-19-related impact, was the females; namely, the group with the highest mean ranks recorded overall.

#### 3.4.2. Age Differences

According to Table 6, a statistically significant difference between age groups was found in terms of all positive wellbeing indices examined; that is, the presence of meaning in life, personal wellbeing, and positive relationships. Through pairwise multiple comparisons, significant differences occurred among all age groups under consideration, except from i. <17 vs. 18–25, 46–55 vs. 56–65, and ≥66 vs. 36–45 for the *presence of meaning in life* scale; ii. <17 vs. 18 to 45 age groups and 46–55 vs. 56 to ≥66 age groups for the *positive relationships* scale. Regarding the *personal wellbeing* scale, statistically significant differences were found only for the age groups of 18–25 years vs. 26 to 65 years.

According to Table 7, a statistically significant difference between age groups was found in terms of the search for meaning in life and COVID-19-related impact. For the *search for meaning in life* scale, post hoc multiple comparisons revealed significant differences among all age groups under consideration, except from the <17 vs. 18 to 65 and 46 to 55 vs. 56 to 65 age groups. Regarding the *COVID-19-related impact*, statistically significant differences were found only for the 18–25 vs. 26–35, 46–55 vs. 18–25, 36–45, 56–65, and ≥66 vs. 18 to 45 age groups.

According to Table 8, a statistically significant difference between age groups was found in terms of depression, anxiety, and stress subscales. Post hoc multiple comparisons revealed significant differences among all age groups under consideration, except from i. <17 vs. 18–25, 26–35 vs. ≥66, and 36–45 vs. 46 to 65 age groups, and 46–55 vs. 56–65 for *depression*; ii. 18–25 vs. ≥66, 36–45 vs. 46 to 65 age groups, and 46–55 vs. 56–65 for *anxiety*; iii. <17 vs. 18–25, 46–55 vs. 36–45, 56–65, and ≥66 vs. 36 to 65 age groups for *stress*.

#### 3.4.3. Marital Status Differences

According to Table 9, a statistically significant difference between marital status groups was found in terms of all positive cognitive–emotional constructs examined; that is, the presence of meaning in life, personal wellbeing, and positive relationships. Through pairwise multiple comparisons, significant differences occurred among all pairs of marital status groups under consideration, except from the divorced and widowed in all scales examined.

According to Table 10, a statistically significant difference between marital status groups was found in terms of the search for meaning in life and COVID-19-related impact. For the *search for meaning in life* scale, post hoc multiple comparisons revealed significant differences among all pairs of marital status groups. Regarding the *COVID-19 Overall Impact Scale*, statistically significant differences were found only for the unmarried compared to those who were divorced, and for those that were married or living together compared with those that were divorced or widowed.

According to Table 11, a statistically significant difference between marital status groups was found in terms of depression, anxiety, and stress subscales. Post hoc multiple comparisons revealed significant differences among all pairs of marital status groups under consideration, except from i. those who were widowed compared to those who were unmarried or divorced for the *depression* subscale; ii. those who were unmarried compared to those who were divorced and widowed and to those who were unmarried or divorced for the *anxiety* subscale; iii. those who were widowed compared to those who were married or living together or divorced for the *stress* subscale.

## 4. Discussion

The purpose of the present study was firstly to investigate and measure the COVID-19-related consequences nationwide, and secondly to examine the possible individual differences from the sociodemographic standpoint on both positive (i.e., personal wellbeing, the presence of meaning in life, and positive relationships) and negative wellbeing indices (namely, depression, anxiety, stress symptoms, and the search for life meaning), but also on the perceived COVID-19-related impact. In greater detail, the study delved into the influence of age, gender, and marital status on the aforementioned variables during the second quarantine period enforced in response to the COVID-19 pandemic. The findings indicated that the COVID-19 Overall Impact Scale was a valid and reliable unidimensional six-item measure of the perceived COVID-19-related consequences; in addition, it was observed that substantial gender, age, and marital status differences existed across all the study variables.

### 4.1. COVID-19-Related Impact and Wellbeing Indices

First, in accordance with the first research hypothesis, a positive correlation was identified between the impact of COVID-19 and negative indices of wellbeing, while a negative correlation was observed with positive wellbeing indices. Specifically, the results indicate that higher levels of perceived COVID-19 impact are associated with increased psychological symptoms and a heightened search for meaning in life, while they are linked to a reduced presence of meaning and personal wellbeing. These findings align with prior research, which reported increased stress and anxiety during the pandemic [21,22]. The lockdown measures further exacerbated the negative impact on mental health, as COVID-19-related stress was more pronounced during the lockdown period than in the subsequent weeks [74]. During the second lockdown in Greece, individuals reported elevated levels of perceived stress, post-traumatic stress disorder symptoms, and loneliness, along with less adaptive coping strategies and lower resilience levels compared to the first lockdown [51,52]. Furthermore, the rates of depression, anxiety, and suicidal thoughts increased during the first lockdown and compared to prepandemic levels, with anxiety levels showing a further surge during the second lockdown [53]. Additionally, the perceived financial and psychological impacts of the COVID-19 pandemic appeared to have a significant influence on individuals’ life satisfaction levels [55].

In terms of the connection between the meaning in life and COVID-19-related consequences, prior research underscored that a self-perceived sense of meaning in life was inversely related to the stress and worries associated with the pandemic [75]. Moreover, during the lockdowns, the presence of meaning in life was notably higher than in the weeks following the lockdown, when mental distress and the quest for meaning in life increased [74]. Attoe and Chimakonam [76] attempted to explain these findings by highlighting that the pandemic’s challenges, such as suffering, isolation, and economic hardships, disrupted individuals’ ability to create meaningful moments in life. Nevertheless, many people managed to find new ways of generating small yet meaningful moments through caring for others or by fostering positive personal and interpersonal meaningful experiences, such as spending quality time with family members [23]. Overall, previous studies emphasized the protective effects of a sense of meaning in life against depression, anxiety, and COVID-19-related stress [27,28].

Regarding the relationship between meaning in life and COVID-19-related consequences, previous research highlighted that the self-perceived presence of meaning in life was negatively associated to stress and worry related to the COVID-19 pandemic [75]. Moreover, during the lockdown, the presence of meaning in life was significantly higher than during the weeks following the lockdown, when mental distress and the search for meaning in life increased [74]. Attoe and Chimakonam [76] attempted to interpret these findings based on the fact that the pandemic increased suffering, isolation, economic hardship, and so forth, and disrupted humanity’s abilities to create moments of meaning in life; simultaneously, many persons found new ways of creating meaningful moments, albeit small, through self-sacrifice/care or by creating positive personal and interpersonal meaningful moments, such as spending quality time with family members [23]. This could be a possible explanation for the fact that the perceived COVID-19-related impact was found to be uncorrelated to the experiences of positive relationships in the present study. On the one hand, individuals reported less perceived support [51], higher levels of loneliness, and feelings of longing for touch, especially those living alone [36,37,38]. However, those living with loved ones perhaps found ways to create a different sense of meaning in life. In general, previous studies underlined the protective effects of the presence of meaning in life against depression, anxiety, and COVID-19-related stress [27,28].

### 4.2. The Effects of Gender, Age, and Marital Status on Wellbeing Indices

#### 4.2.1. Wellbeing of Men and Women

Regarding gender differences, the findings of the present study suggest that men tended to express greater satisfaction with their lives during lockdown, while women experienced heightened levels of depression, anxiety, stress symptoms, a more pronounced perceived impact of COVID-19 on their lives, and a stronger quest for meaning in life. The sole positive well-being aspect where women surpassed men was in the realm of positive interpersonal relationships. Additionally, there were no gender differences identified in levels of the presence of meaning in life. These results partially support the second research hypothesis, which anticipated that male participants would report significantly higher positive well-being indices, fewer psychological symptoms, and lower COVID-19 impact compared to female participants.

Prepandemic literature indicated that women typically reported higher levels of depression and anxiety [59,77]. However, previous research on gender differences in overall wellbeing levels has yielded inconsistent results. In prior adversities, such as the Greek economic crisis, women reported significantly higher levels of depression, anxiety, stress, the search for meaning, and perceived economic crisis consequences. Nevertheless, no significant gender differences were found in the presence of meaning in life [25].

During the COVID-19 pandemic, research findings in Greece have affirmed that women reported higher levels of psychological symptoms and a greater psychological impact during the second wave of the pandemic compared to men [54,55], and lower levels of life satisfaction [47]. Despite a higher fatality rate for men compared to women, the COVID-19 pandemic had a more significant psychological impact on women [78].

Several explanations exist for these findings. Firstly, most of the healthcare professionals are women, and such professions have been associated with a higher likelihood of experiencing symptoms of depression, anxiety, insomnia, post-traumatic stress, and psychological distress during the pandemic [45,79]. Additionally, women who were pregnant, in the postpartum period, experienced a miscarriage, or facing intimate partner violence were at a particularly high risk of developing mental health issues during the pandemic [80]. The perinatal period is a time of heightened vulnerability to mental health problems, with approximately one in seven perinatal women experiencing increased anxiety, depression, and distress, especially those with medically high-risk pregnancies [81]. The maternal role also contributed to women’s stress, as during the second Greek lockdown, schools were closed for seven months, and mothers typically shouldered the bulk of childcare and eldercare responsibilities in Greece. Consequently, they reported high levels of depression and anxiety [82].

Lastly, for many women, the home, which they were advised to stay in for safety, represented the least safe place, as they were often victims of intimate partner violence. This phenomenon tripled during the COVID-19 lockdowns [83]. Various factors, such as financial problems, social isolation, loss of support systems, confined living spaces, the loss of loved ones, fear of death, difficulties in accessing medical and social services, the inability to leave, and increased consumption of addictive substances, contribute to the risk of intimate partner violence, particularly in crisis contexts, when male aggression tends to escalate [78]. All of these factors help to explain why women’s wellbeing was more precarious during the COVID-19 pandemic. On the other hand, research underscores the protective role of perceived social support on women’s mental health [78,80], and women tended to report higher levels of social support than men [84].

#### 4.2.2. Age Differences on Wellbeing

According to our third research hypothesis, we expected that middle-aged and older adults would report significantly higher positive wellbeing indicators, fewer psychological symptoms, and less COVID-19 impact compared to young adults. This hypothesis was substantiated, with one exception: the variable related to perceived positive relationships, where young adults reported higher levels than their middle-aged and older counterparts.

Indeed, previous research findings suggest that, during the lockdown, older adults adapted more easily, experienced less stress related to the pandemic, fewer disruptions in their lives, less social isolation, higher life satisfaction, and better quality in their close relationships compared to younger individuals [3,4,46,47]. In contrast, young adults, particularly during the second Greek lockdown, faced challenges, such as reduced levels of positive emotions and life satisfaction, increased symptoms of depression, anxiety, and stress [43,54,55].

The lives of young adults in Greece underwent significant changes during the COVID-19 quarantine, with schools and universities shifting to emergency remote teaching. This change amplified academic stressors, particularly for university students who were dealing with an unprecedented level of disruption and uncertainty [85]. Simultaneously, emerging adults, who are in the process of exploring their self-identity and sometimes find themselves in a transitional phase between adolescence and adulthood [86], encountered difficulties in pursuing their normal lives, finding employment, or socializing. Many young adults reported struggling with deteriorating mental health during the pandemic, a lack of support, and concerns about re-establishing social connections after the pandemic. However, some young adults described their experiences during the pandemic as beneficial, citing increased awareness of mental health and greater comfort in discussing it, as well as stronger bonds with family members [87]. This could explain why young adults reported higher levels of perceived positive relationships.

#### 4.2.3. Together or Alone during the Quarantine? Implications for Adult Wellbeing

As per the fourth hypothesis in our study, we anticipated that participants who were married or living with a romantic partner would report significantly higher positive wellbeing indicators, fewer psychological symptoms, and less COVID-19 impact compared to those who were single, widowed, or divorced. The findings only partially supported this hypothesis, with the unexpected discovery that unmarried participants reported higher levels of perceived positive relationships compared to their married or cohabiting counterparts.

In Greece, unmarried individuals primarily consist of young adults. Therefore, the earlier explanations regarding the fragile mental health during the developmental phase of emerging adulthood, life changes due to increased unemployment, restrictions on socialization, and the transition to emergency remote teaching in universities help clarify the elevated levels of psychological symptoms, the search for meaning, and the lower overall wellbeing in this group. 

On the other hand, previous studies emphasized the protective role of a supportive family environment for individuals’ mental health during the pandemic [7]. Furthermore, married individuals over the age of 40 were found to adapt more readily to the unprecedented quarantine conditions [46]. People living with significant others discovered new ways to create meaningful moments together; for example, married Australians reported higher levels of resilience during the lockdown due to positive changes in family dynamics, such as spending quality time with family members [23].

Additionally, the presence of family or living with significant others frequently served as a protective system against the adverse effects of the COVID-19 pandemic. Several factors contributed to this, including perceived social support, a sense of belonging, social connectedness, reduced loneliness, shared family beliefs, and positive family communication. Specifically, research findings indicated that feeling connected to family members positively predicted wellbeing and negatively predicted perceived stress during the lockdown [88]. Furthermore, staying at home with close relatives enhanced social connectedness during the lockdown period, which, in turn, was associated with lower levels of perceived stress, fatigue, as well as general and COVID-19-specific concerns [89]. Another beneficial aspect of living with family is the increased perception of family support, which negatively predicts depression levels [90] and reduces feelings of loneliness [91].

Moreover, systemic models suggest that families function as systems that strive to mitigate the extent and type of disruption occurring during times of adversity [92]. Two family processes supported this phenomenon: (a) the establishment and maintenance of positive family relationships that offset the challenges of otherwise distressing situations, and (b) the adaptation of family belief systems to provide a framework for a better understanding of the stressors related to the COVID-19 pandemic [93]. These processes enhanced both family and individual resilience, as well as overall wellbeing [94]. Additionally, daily practices of gratitude, effective communication, and engaging in positive activities together contributed to an increased sense of togetherness, trust, and cohesion [95]. This explains why the literature indicates that most parents reported perceiving more positive changes during the COVID-19 pandemic than negative ones, particularly in terms of feeling emotionally closer to their children and spending more enjoyable time with them [96]. 

### 4.3. Limitations and Recommendations for Future Studies

The present study was a one-time, self-report survey that exclusively examined constructs through Likert-type scales. As a result, participants’ responses might have been influenced by various response biases, such as the desire to present oneself favorably. Therefore, conducting more comprehensive research with innovative and ongoing data collection methods, such as daily diary surveys and ecological momentary assessments extending beyond the initial stages of the pandemic, could offer a deeper understanding of these findings and the underlying dynamic processes. The publication of annual Greek Wellbeing Reports, which utilize both longitudinal and cross-sectional data, will provide additional insights into the long-term effects of the COVID-19 pandemic on individuals with diverse demographic characteristics. This approach will enable comparisons of the effects of various types of crises, such as floods and earthquakes, and create opportunities for the development and implementation of customized interventions aimed at preventing mental illness and promoting mental health and overall wellbeing.

## 5. Conclusions

The current study made a valuable addition to the body of research by introducing a measure of COVID-19-related effects using a broad, nationwide sample from Greece. It also offered insights into the associations between COVID-19-related impact and wellbeing indices. Finally, it shed light on the variations in wellbeing indicators related to gender, age, and marital status, specifically during the extended seven-month lockdown in Greece. These findings can serve as a valuable guide for policymakers and mental health experts to develop more personalized and inclusive support systems for women, young adults, and unmarried individuals during any future adversities.

## Figures and Tables

**Table 1 ijerph-20-07182-t001:** Descriptive statistics of the COVID-19 overall impact scale (*n* = 16,906).

Item	Mean (SD)	r_tot_	α if Item Deleted
Item 1	5.65 (3.08)	0.38	0.82
Item 2	7.96 (2.31)	0.54	0.78
Item 3	7.92 (2.35)	0.61	0.76
Item 4	6.92 (2.68)	0.70	0.74
Item 5	7.43 (2.48)	0.64	0.75
Item 6	6.06 (3.06)	0.54	0.78

Note: r_tot_  =  corrected item–total correlation, Cronbach’s alpha = 0.80.

**Table 2 ijerph-20-07182-t002:** Descriptive analysis results of positive/negative wellbeing indices and COVID-19-related impact constructs (*n* = 16,906).

Variables	Mean (SD)	Median	Mode
Presence of meaning in life	7.22 (1.72)	7.40	8.00
Personal wellbeing	7.18 (1.28)	7.38	8.00
Positive relationships with others	8.04 (1.44)	8.26	10.00
Search for meaning in life	7.14 (1.89)	7.40	8.00
Depression	3.87 (2.22)	3.43	2.00
Anxiety	3.41 (2.21)	2.71	1.00
COVID-19 overall impact	6.64 (1.83)	6.71	7.00

Note: minimum value = 1, maximum value = 10.

**Table 3 ijerph-20-07182-t003:** Factor loading and communality of the COVID-19 overall impact scale items (*n* = 16,906).

Item	Loading	r_tot_
1. The pandemic has financial impacts on my life.	0.41	0.17
2. The pandemic changed my daily routine.	0.61	0.37
3. I am annoyed with the changes having occurred in my life due to the pandemic.	0.71	0.51
4. The pandemic has psychological impacts on my life.	0.82	0.66
5. The pandemic has social impacts on my life.	0.74	0.55
6. The pandemic has physical impacts on my life.	0.60	0.37

**Table 4 ijerph-20-07182-t004:** Mann–Whitney U-test values of positive wellbeing indices in terms of gender (*n* = 16,906).

Variable	Gender	Mean Rank	U-Value ^a^ (z Statistic)	*p* ^b^
Presence of meaning in life	Male (*n* = 5848)	8504.30	32,018,954.00 (−1.01)	0.310
	Female (*n* = 11,055)	8424.33		
Personal wellbeing	Male (*n* = 5848)	8689.89	30,933,640.50 (−4.61)	<0.001
	Female (*n* = 11,055)	8326.16		
Positive relationships with others	Male (*n* = 5848)	7741.03	28,167,086.00 (−13.78)	<0.001
	Female (*n* = 11,055)	8828.10		

Note: ^a^ adjusted for ties; ^b^ two-tailed.

**Table 5 ijerph-20-07182-t005:** Mann–Whitney U-test values of negative wellbeing indices in terms of gender (*n* = 16,906).

Variable	Gender	Mean Rank	U-Value ^a^ (z Statistic)	*p* ^b^
Search for meaning in life	Male (*n* = 5848)	8298.97	31,429,919.50 (−2.97)	0.003
	Female (*n* = 11,055)	8532.95		
Depression	Male (*n* = 5848)	7969.83	29,505,104.50 (−9.35)	<0.001
	Female (*n* = 11,055)	8707.06		
Anxiety	Male (*n* = 5848)	7896.62	29,076,943.50 (−10.77)	<0.001
	Female (*n* = 11,055)	8745.79		
Stress	Male (*n* = 5848)	7864.76	28,890,641.00 (−11.38)	<0.001
	Female (*n* = 11,055)	8762.64		
COVID-19 overall impact	Male (*n* = 5848)	7843.68	28,767,374.50 (−11.79)	<0.001
	Female (*n* = 11,055)	8773.80		

Note: ^a^ adjusted for ties; ^b^ two-tailed.

**Table 6 ijerph-20-07182-t006:** Κruskal–Walls H-test values of positive wellbeing indices in terms of age groups (*n* = 16,906).

Variable	Age Group	Mean Rank	*H* ^a^	*p* ^b^
Presence of meaning in life	<17 (*n* = 115)	6457.62	1113.633	<0.001
	18–25 (*n* = 4974)	6866.53		
	26–35 (*n* = 3312)	7901.55		
	36–45 (*n* = 2681)	9263.39		
	46–55 (*n* = 3598)	9855.32		
	56–65 (*n* = 1240)	10,074.16		
	≥66 (*n* = 970)	9063.13		
Personal wellbeing	<17 (*n* = 115)	8156.47	60.196	<0.001
	18–25 (*n* = 4974)	8057.48		
	26–35 (*n* = 3312)	8460.71		
	36–45 (*n* = 2681)	8792.23		
	46–55 (*n* = 3598)	8634.84		
	56–65 (*n* = 1240)	8833.02		
	≥66 (*n* = 970)	8261.51		
Positive relationships with others	<17 (*n* = 115)	9225.55	608.425	<0.001
	18–25 (*n* = 4974)	9541.32		
	26–35 (*n* = 3312)	8882.90		
	36–45 (*n* = 2681)	8297.90		
	46–55 (*n* = 3598)	7470.80		
	56–65 (*n* = 1240)	7056.79		
	≥66 (*n* = 970)	7039.04		

Note: ^a^ adjusted for ties; ^b^ two-tailed; df = 6.

**Table 7 ijerph-20-07182-t007:** Κruskal–Walls H-test values of the search for meaning in life and COVID-19-related impact in terms of age groups (*n* = 16,906).

Variable	Age Group	Mean Rank	*H* ^a^	*p* ^b^
Search for meaning in life	<17 (*n* = 115)	9131.60	282.231	<0.001
	18–25 (*n* = 4974)	9145.62		
	26–35 (*n* = 3312)	8720.92		
	36–45 (*n* = 2681)	8244.63		
	46–55 (*n* = 3598)	7897.46		
	56–65 (*n* = 1240)	8159.30		
	≥66 (*n* = 970)	6787.51		
COVID-19 overall impact	<17 (*n* = 115)	8836.83	92.749	<0.001
	18–25 (*n* = 4974)	8843.98		
	26–35 (*n* = 3312)	8454.29		
	36–45 (*n* = 2681)	8632.72		
	46–55 (*n* = 3598)	8139.99		
	56–65 (*n* = 1240)	7751.23		
	≥66 (*n* = 970)	7829.00		

Note: ^a^ adjusted for ties; ^b^ two-tailed; df = 6.

**Table 8 ijerph-20-07182-t008:** Κruskal–Walls H-test values of depression, anxiety, and stress symptoms in terms of age groups (*n* = 16,906).

Variable	Age Group	Mean Rank	*H* ^a^	*p* ^b^
Depression	<17 (*n* = 115)	10,637.21	424.773	<0.001
	18–25 (*n* = 4974)	9473.13		
	26–35 (*n* = 3312)	8500.91		
	36–45 (*n* = 2681)	7739.73		
	46–55 (*n* = 3598)	7744.03		
	56–65 (*n* = 1240)	7475.68		
	≥66 (*n* = 970)	8519.38		
Anxiety	<17 (*n* = 115)	10,743.03	354.073	<0.001
	18–25 (*n* = 4974)	9328.27		
	26–35 (*n* = 3312)	8393.99		
	36–45 (*n* = 2681)	7853.59		
	46–55 (*n* = 3598)	7802.40		
	56–65 (*n* = 1240)	7494.91		
	≥66 (*n* = 970)	9058.93		
	<17 (*n* = 115)	10,430.81		
	18–25 (*n* = 4974)	9497.23		
	26–35 (*n* = 3312)	8635.72		
Stress	36–45 (*n* = 2681)	7961.64	456.346	<0.001
	46–55 (*n* = 3598)	7703.33		
	56–65 (*n* = 1240)	7276.47		
	≥66 (*n* = 970)	7752.25		

Note: ^a^ adjusted for ties; ^b^ two-tailed; df = 6.

**Table 9 ijerph-20-07182-t009:** Κruskal–Walls H-test values of positive wellbeing indices in terms of marital status (*n* = 16,906).

Variable	Age Group	Mean Rank	*H* ^a^	*p* ^b^
Presence of meaning in life	Unmarried (*n* = 7988)	7084.96	1222.810	<0.001
	Married/living together (*n* = 7503)	9803.11		
	Divorced (*n* = 943)	9032.61		
	Widowed (*n* = 472)	9003.61		
Personal wellbeing	Unmarried (*n* = 7988)	7946.93	360.415	<0.001
	Married/living together (*n* = 7503)	9222.17		
	Divorced (*n* = 943)	7236.68		
	Widowed (*n* = 472)	7238.69		
Positive relationships with others	Unmarried (*n* = 7988)	9059.39	291.555	<0.001
	Married/living together (*n* = 7503)	8080.33		
	Divorced (*n* = 943)	7106.76		
	Widowed (*n* = 472)	6822.05		

Note: ^a^ adjusted for ties; ^b^ two-tailed; df = 6.

**Table 10 ijerph-20-07182-t010:** Κruskal–Walls H-test values of search for meaning in life and COVID-19-related impact in terms of marital status (*n* = 16,906).

Variable	Age Group	Mean Rank	*H* ^a^	*p* ^b^
Search for meaning in life	Unmarried (*n* = 7988)	8999.45	247.935	<0.001
	Married/living together (*n* = 7503)	7995.13		
	Divorced (*n* = 943)	8490.29		
	Widowed (*n* = 472)	6426.80		
COVID-19 overall impact	Unmarried (*n* = 7988)	8673.91	38.252	<0.001
	Married/living together (*n* = 7503)	8242.16		
	Divorced (*n* = 943)	8572.01		
	Widowed (*n* = 472)	7845.98		

Note: ^a^ adjusted for ties; ^b^ two-tailed; df = 6.

**Table 11 ijerph-20-07182-t011:** Κruskal–Walls H-test values of depression, anxiety, and stress symptoms in terms of marital status (*n* = 16,906).

Variable	Age Group	Mean Rank	*H* ^a^	*p* ^b^
Depression	Unmarried (*n* = 7988)	9202.63	423.122	<0.001
	Married/living together (*n* = 7503)	7602.25		
	Divorced (*n* = 943)	8622.24		
	Widowed (*n* = 472)	8970.01		
Anxiety	Unmarried (*n* = 7988)	8968.56	229.656	<0.001
	Married/living together (*n* = 7503)	7822.91		
	Divorced (*n* = 943)	8711.01		
	Widowed (*n* = 472)	9246.34		
Stress	Unmarried (*n* = 7988)	9193.30	366.719	<0.001
	Married/living together (*n* = 7503)	7700.76		
	Divorced (*n* = 943)	8414.56		
	Widowed (*n* = 472)	7976.85		

Note: ^a^ adjusted for ties; ^b^ two-tailed; df = 6.

## Data Availability

The data presented in this study are available on request from the corresponding author.

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
