# Peer review of "Individual Differences on Wellbeing Indices during the COVID-19 Quarantine in Greece: A National Study"

_ijerph, 2023, doi:10.3390/ijerph20247182_

Round 1

Reviewer 1 Report

Comments and Suggestions for Authors

Thank you for the opportunity to review this manuscript.  This research is timely and of interest to multiple professions in the post-Covid19 pandemic world. Please see below for edit recommendations:

Line 51 collapse of healthcare systems.

This is a thorough and extensive study well worth publishing.  Your self-developed scale, the CV19OIS is well created and I am impressed with your focus on a multitude of dynamics at play in wellness.  Meaning of life is a widely experienced phenomenon, present in culture worldwide.  Compared to examining mental health experiences alone (depression, anxiety) you have examined a factor that is culturally relevant. 

Very well done.  With the above edit I feel this study is ready for publication.

Author Response

Thank you very much for your positive feedback on our manuscript.

We have incorporated the change based on your comment regarding Line 51.

Reviewer 2 Report

Comments and Suggestions for Authors

The impact of COVID-19 and its associated lockdown phases on people's physical and mental health, as well as their daily lives and functioning, is an important focus of research. The present study aimed to investigate the consequences associated with COVID-19 across the country from a sociodemographic point of view. An undoubted advantage of the work is the large study group. In my opinion, the work still needs to be supplemented with some information

My doubts are raised by the first research question

RQ1: Is a short measure of the perceived COVID-19 impact unidimensional, conceptually valid and reliable?

It would be useful to first consider whether the measure is theoretically unidimensional or multidimensional and only verify this. My doubt is confirmed by the lack of a hypothesis for this research question. These deficiencies should be remedied.
In my opinion, it is worth including even a brief justification of the hypothesis under each hypothesis

The study included 16,906 Greeks from all age groups. However, in the description of the group, the authors do not comment on the distribution of the control variables (e.g. gender, education) in relation to the population. Given that the data are related to the population, it is worthwhile for each variable describing the group to refer to the distribution of this variable in the study population

LINE 289-296
By whom was the scale adopted into Greek ? Was it done as part of this project - please specify. The literature cited by the authors refers to a different scale ....

LINE 311 and what was the cronbach's alpha for the subscale ?

Author Response

Thank you very much for your helpful comments. We appreciate that your comments assisted us to make constructive modifications and we hope that the revised paper adequately address the issues raised by you. Following your suggestions, we approached some parts of the Method section from a different standpoint to better support our study presentation, and we made several additions and clarifications across the text, as requested. 

Below, we have provided our responses that describe in detail our efforts to address all the changes requested by you, with pointers to relevant sections in the paper where these have been addressed.

Comment 1:

It would be useful to first consider whether the measure is theoretically unidimensional or multidimensional and only verify this. My doubt is confirmed by the lack of a hypothesis for this research question. These deficiencies should be remedied. In my opinion, it is worth including even a brief justification of the hypothesis under each hypothesis.

Answer to comment 1:

Thank you very much for pointing this out. First of all, in the Method section of the manuscript we have added information regarding previous models and theories that support our rationale regarding the COVID-19 indicators included in the scale. These models support that these indicators can be measured separately but also provide information about a latent higher-order variable, which is labeled quality of life, wellbeing, and crisis consequences, respectively. Secondly, based on your valuable input, we included a brief justification of each hypothesis.

Comment 2:

The study included 16,906 Greeks from all age groups. However, in the description of the group, the authors do not comment on the distribution of the control variables (e.g. gender, education) in relation to the population. Given that the data are related to the population, it is worthwhile for each variable describing the group to refer to the distribution of this variable in the study population.

Answer to comment 2:

We greatly appreciate your contribution, which enhances the quality of our manuscript. Following your feedback, we incorporated information from the Hellenic Statistical Authority concerning gender, marital status, educational level, and age distribution within the general population in Greece. Additionally, we provided commentary on how the characteristics of our sample compare to those of the general Greek population.

Comment 3:

LINE 289-296

By whom was the scale adopted into Greek? Was it done as part of this project - please specify. The literature cited by the authors refers to a different scale ....

Answer to comment 3:

Thank you for bringing this to our attention. In response to your feedback, we have included information stating that the scale was adapted by the authors specifically for this project. Furthermore, the citation used pertains to a chapter that details the steps involved in adapting psychometric instruments to different cultural contexts. Please see below:

Pezirkianidis, C., Karakasidou, E., Dimitriadou, D., & Stalikas, A. (2017). Translation and adaptation of psychometric instruments. In M. Galanakis, C. Pezirkianidis, & A. Stalikas (Eds), Basic aspects of psychometrics, 489-512. Topos.

Comment 4:

LINE 311 and what was the cronbach's alpha for the subscale?

Answer to comment 4:

Thank you for your comment. We did not include the Cronbach’s alpha coefficients for each subscale, since we didn’t use them in this manuscript. Based on your kind comment, we have added Cronbach's alpha for the subscales.

Reviewer 3 Report

Comments and Suggestions for Authors

The paper addresses a pertinent issue - individual differences in wellbeing during the COVID-19 quarantine in Greece. The study adds valuable insights to the understanding of the impact of the pandemic on mental health and overall, I consider that the paper is well-structured and contributes to the existing literature.

I think that the methods section could be completed with some information, for example, was any calculation or effort made to ensure that the sample was representative of the Greek population? How?Otherwise, nothing to report.

Author Response

We greatly appreciate your contribution, which enhances the quality of our manuscript.

Following your feedback, we incorporated information from the Hellenic Statistical Authority concerning gender, marital status, educational level, and age distribution within the general population in Greece. Additionally, we provided commentary on how the characteristics of our sample compare to those of the general Greek population.